# Comparison of SARS-CoV-2 Detection from Saliva Sampling and Oropharyngeal Swab

Mia de Laurent Clemmensen,[a] Kamilla Kolding Bendixen,[a] Katharina Flugt,[a] Pernille Pilgaard,[a] Ulf Bech Christensen[a]

[a]PentaBase A/S, Odense, Denmark

**ABSTRACT** We examined the detection rate of severe acute respiratory syndrome coronavirus 2 (SARS-CoV-2) using reverse transcription-PCR (RT-PCR) of side-by-side saliva and oropharyngeal swab (OPS) samples from 639 symptomatic and asymptomatic subjects, of which 47 subjects were found to be positive for SARS-CoV-2 in the OPS or saliva sample or both. It was found that the detection rate (93.6% for both OPS and saliva) as well as the sensitivity and specificity were comparable between the two sampling methods in this cohort. The sensitivity was 0.932 (95% confidence interval [CI], 0.818 to 0.977) and the specificity was 0.995 (95% CI, 0.985 to 0.998), both for saliva when OPS sampling was used as the reference and for OPS when saliva was used as the reference. Furthermore, the Cohen's kappa value was 0.926 (95% CI, 0.868 to 0.985), indicating strong agreement between the two sampling methods. In addition, the viral RNA stability in pure saliva and saliva mixed with preservation buffers was examined following storage at room temperature and at 4°C. It was found that pure saliva kept the viral RNA stable for 9 days at both temperatures and that the type of preservation buffer can either enhance or reduce the stability of the RNA. We conclude that self-administered saliva sampling is an attractive alternative to oropharyngeal swabbing for SARS-CoV-2 detection, and it might be useful in large-scale testing.

**IMPORTANCE** It is not inconceivable that we will witness recurring surges of COVID-19 before the pandemic finally recedes. It is therefore still relevant to look for feasible, simple, and flexible screening methods so that schools, workplaces, and communities in general can avoid lockdowns. In this work, we analyzed two different sampling methods: oropharyngeal swabs and saliva collection. Oropharyngeal swabs must be collected by trained health personnel at clinics, whereas self-assisted saliva collection can be performed at any given location. It was found that the two sampling methods were comparable. Saliva sampling is a simple method that allows easy mass testing using minimal resources from the existing health care system, and this method may therefore prove to be an effective tool for containing the COVID-19 pandemic.

**KEYWORDS** COVID-19, oropharyngeal swab, RNA stability, RT-qPCR, SARS-CoV-2, saliva

Severe acute respiratory syndrome coronavirus 2 (SARS-CoV-2) caused the outbreak of coronavirus disease 2019 (COVID-19) in December 2019. The World Health Organization (WHO) declaimed it as a pandemic on 11 March 2020 (1), and the disease has amounted to more than 486 million cases and 6.1 million deaths as of 1 April 2022 (https://covid19.who.int/). Infection with SARS-CoV-2 can lead to severe illness with lethal respiratory difficulties, and the virus is especially dangerous and difficult to control as it can infect individuals asymptomatically, leading to unconscious spread of the virus. Several vaccines have been developed, with Pfizer-BioNTech being one of the most frequently used vaccines. Primary efficacy analysis of the Pfizer-BioNTech vaccine has reported it to be 95% effective against symptomatic SARS-CoV-2 infection (2) (ClinicalTrials.gov identifier NCT04368728). Unfortunately, waning immunity from the vaccine has been reported 6 months postvaccination (3).

Address correspondence to Mia de Laurent Clemmensen, miadelaurent@gmail.com.

The authors declare a conflict of interest. During the study, the authors worked at PentaBase A/S, Denmark, who supply RT-qPCR assays for detection of SARS-CoV-2 in OPS, NPS, or saliva samples.

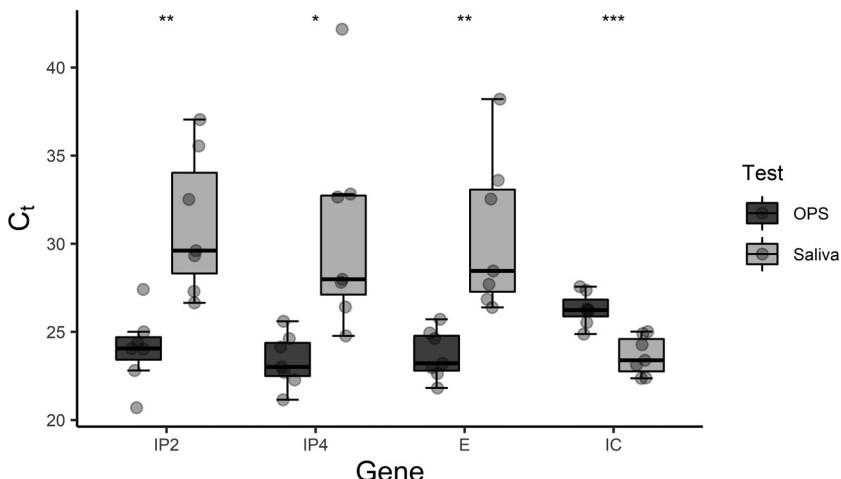

**FIG 1** Boxplot of $C_T$ values of the three SARS-CoV-2-specific genes (IP2, IP4, and E) and an internal control (IC) for $n = 7$ SARS-CoV-2-positive samples. The data are grouped by sampling method, oropharyngeal swab (OPS) or saliva, with buffer 1 used for preservation of the saliva. It is evident that the mean $C_T$ values were significantly higher for saliva samples than for oropharyngeal swab samples for the virus-specific genes (an unpaired $t$ test was used). See data in Table 1 posted at https://5ba .se/MLCetal/Saliva. ns, $P > 0.05$; *, $P \leq 0.05$; **, $P \leq 0.01$; ***, $P \leq 0.001$; ****, $P \leq 0.0001$.

Testing for SARS-CoV-2 in asymptomatic and symptomatic individuals has been a tool for tracking and controlling the pandemic. Oropharyngeal swabs (OPS) and nasopharyngeal swabs (NPS) are the standard sampling procedure methods, but both methods are invasive, cause discomfort, and require trained personnel. Furthermore, the personnel are at risk due to short physical distances and exposure to aerosols when performing the swabbing (4). Saliva sampling has gained more interest, as it is an easy, painless, and noninvasive sampling form. In addition, saliva tests can be self-administered, thereby decreasing the risk to and need for health care personnel, as well as the risk of transmitting the infection to other persons getting tested at the site. This sampling method provides better opportunities for simple, inexpensive, flexible, and accessible large-scale testing (5, 6). The worldwide emergence of Omicron (B.1.1.529) subvariants has resulted in record-breaking spread and infection rates, which emphasizes that the pandemic is not over and that we can still benefit from large-scale testing (7, 8). Several studies have shown that self-administered saliva sampling can easily be used in large-scale screening of SARS-CoV-2 in asymptomatic individuals, while being cost-efficient and reducing the risk of transmitting the virus from infected individuals to health care personnel (9–13). A study by Yokota et al. (9) was performed in Japan, which since has introduced saliva as a valid sampling method for nucleic acid amplification testing that meets the requirements to enter the country (14). In addition, a technical report by the European Centre for Disease Prevention and Control (ECDC) in May 2021 collected evidence that saliva can be used in symptomatic patients and for repeated screening of asymptomatic individuals (15).

This study investigated viral RNA stability in saliva samples mixed with different preservation media or when kept as pure saliva. Furthermore, this study aimed to compare OPS to saliva sampling using reverse transcription-quantitative real-time PCR (RT-qPCR) to detect SARS-CoV-2.

## RESULTS

**SARS-CoV-2 RNA preservation in saliva using buffer 1.** Preservation buffer 1 was used for preservation of the saliva samples for comparison with OPS. The mean threshold cycle ($C_T$) values for SARS-CoV-2-specific genes (two regions from the RNA-dependent RNA polymerase gene, IP2 and IP4, and one region from the envelope protein gene, E) were significantly lower using the OPS sampling method than using saliva sampling with preservation buffer 1 (IP2, $P = 0.0025$; IP4, $P = 0.016$; and E, $P = 0.0054$), indicating insufficient preservation of RNA from saliva sampling in buffer 1 (Fig. 1). The mean $C_T$ value for the saliva

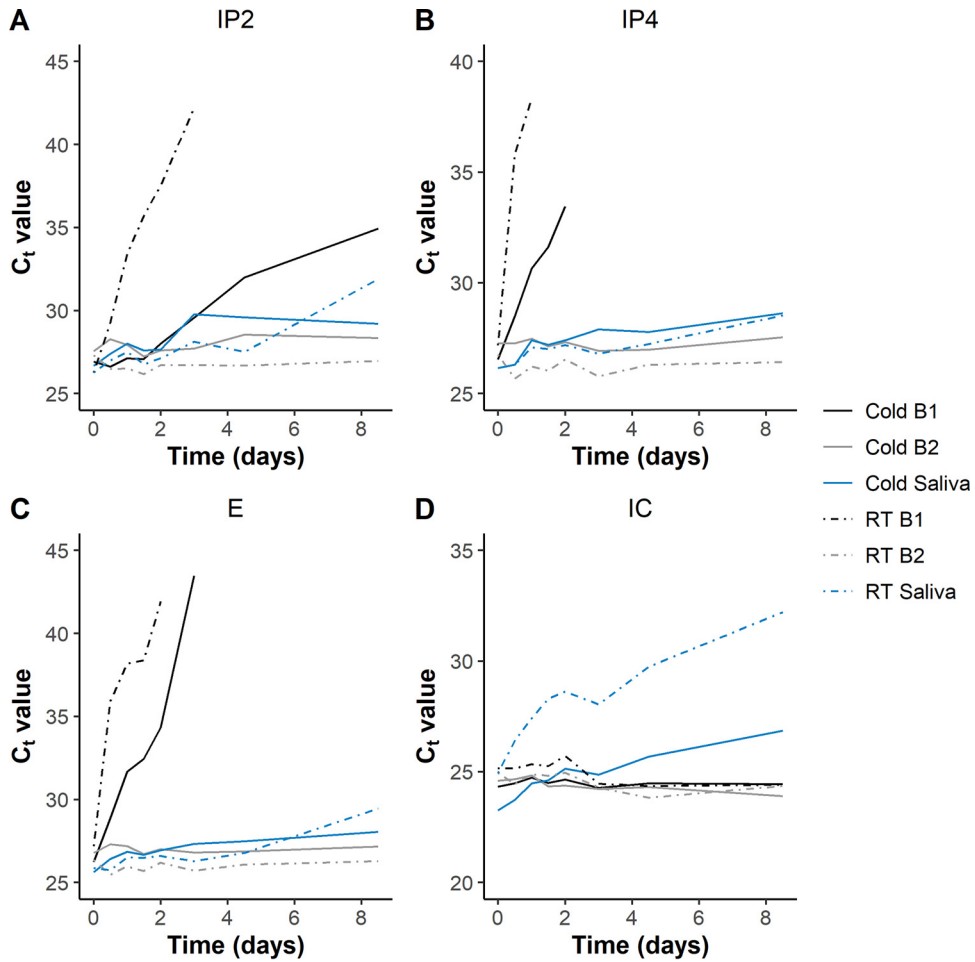

**FIG 2** Stability of SARS-CoV-2-positive saliva as pure saliva (saliva) and in buffer 1 (B1) and buffer 2 (B2). The samples were stored in the refrigerator at 4°C (cold) and at room temperature (RT). The samples were stored over 9 days, and the CoviDetect FAST COVID-19 multiplex RT-qPCR assay was used for analysis of the samples during the period. (A to C) SARS-CoV-2-specific genes IP2, IP4, and E; (D) internal control (IC), the human RNP gene. It is evident that RNA stability in buffer 1 decreased over time both at RT and 4°C, whereas buffer 2 kept the RNA stable. See data in Table 2 posted at https://5ba.se/MLCetal/Saliva.

sampling method was lower on the internal control (IC) gene, indicating better preservation of DNA than RNA in saliva in buffer 1 ($P = 0.00045$).

**RNA stability in saliva and preservation media.** RNA stability was examined in pure saliva and saliva mixed with different preservation buffers (buffers 1 or 2) following storage at room temperature (RT) and 4°C. As seen in Fig. 2, pure saliva showed high RNA stability regardless of the temperature. The $C_T$ values for the three SARS-CoV-2-specific genes were increased on average by 2.5 cycles ($\Delta C_T = 2.5$, $\Delta C_T = 2.5$, and $\Delta C_T = 2.4$ for IP2, IP4, and E, respectively) for pure saliva stored at 4°C and increased on average by 3.9 cycles ($\Delta C_T = 5.6$, $\Delta C_T = 2.4$, and $\Delta C_T = 3.6$ for IP2, IP4, and E, respectively) following storage at RT for 9 days. However, the IC $C_T$ values for human DNA increased more than those for the SARS-CoV-2 RNA sequences, indicating low DNA stability, especially at RT, for pure saliva samples ($\Delta C_T = 7.3$). The inverse was found for the saliva samples stored in both types of preservation buffers, where the $C_T$ values of the IC were unaltered, indicating high DNA stability.

RNA stability differed considerably depending on the preservation buffer used. Following storage of saliva for 3 days in buffer 1 at RT, it was no longer possible to detect any of the SARS-CoV-2-specific genes. In addition, it was not possible to detect the IP4 or E genes in saliva samples preserved with buffer 1 at 4°C following day 2; however, the IP2 gene was detectable during all 9 days.

**TABLE 1** Confusion matrix of the SARS-CoV-2-positive, -negative, and invalid samples for OPS and saliva in buffer 2

| Oropharyngeal swab result | Saliva sample result | | | |
|---|---|---|---|---|
| | Positive | Negative | Invalid | Total |
| Positive | 41 | 3 | 0 | 44 |
| Negative | 3 | 589 | 3 | 595 |
| Invalid | 0 | 0 | 0 | 0 |
| Total | 44 | 592 | 3 | 639 |

Preservation buffer 2 was shown to be a better choice for ensuring high RNA stability over a period of 9 days; the $C_T$ values of both the three viral genes and the IC gene were almost unaffected following storage both at RT and at 4°C.

**Comparison of oropharyngeal swab and saliva sampling using buffer 2.** Following testing of the 639 OPS and 639 saliva samples, 7.4% (47/639) of the participants were found to be positive for SARS-CoV-2 using the OPS or saliva samples or both (Table 1). Virus was detected using both types of sampling methods for 87.2% (41/47) of the positive participants. Among the participants testing positive with discordant results, 6.4% (3/47) only tested positive using the saliva sample, and 6.4% (3/47) only tested positive using the OPS sample. Therefore, there was no difference in the detection rate of SARS-CoV-2 between OPS and saliva sampling in this cohort (OPS, 93.6% [44/47]; saliva, 93.6% [44/47]). When the OPS swab was used as the reference, the sensitivity of the saliva sampling method was 0.932 (95% confidence interval [CI], 0.818 to 0.977), and the specificity was 0.995 (95% CI, 0.985 to 0.998), whereas when the saliva samples were used as the reference, the sensitivity of the OPS sampling method was 0.932 (95% CI, 0.818 to 0.977), and the specificity was 0.995 (95% CI, 0.985 to 0.998).

We observed similar results for the mean $C_T$ values using saliva sampling compared to OPS for the three viral genes (IP2, $P = 0.86$; IP4, $P = 0.7$; and E, $P = 0.5$) (Fig. 3). We compared the IC for the negative and positive samples for the two sampling methods using two-way analysis of variance (ANOVA), and a significant interaction was found between the sampling method and the $C_T$ value of the IC ($P = 0.013$). Using Tukey's honestly significant difference (HSD) test, a difference in the $C_T$ values for the IC was found between the patients testing negative and positive using the saliva method, with the IC $C_T$ values for the positive saliva samples being significantly lower than those for the negative samples ($P = 0.032$). This difference was not observed for the OPS sampling method ($P = 0.87$). Representation of the IC data can be found in Fig. 1 posted at https://5ba.se/MLCetal/Saliva.

The Cohen's kappa value was 0.926 (95% CI, 0.868 to 0.985), indicating strong agreement between the two sampling methods (Table 1). In addition, the average $C_T$ values of the viral genes combined were similar, 29.5 (interquartile range [IQR], 23.9 to 31.8) for saliva sampling and 29.3 (IQR, 23.5 to 36.0) for OPS sampling.

Three samples were found to be SARS-CoV-2 negative using the OPS swabs but invalid using the saliva sampling method (Table 1).

## DISCUSSION

We found that detection of SARS-CoV-2 was highly comparable between saliva sampling and OPS sampling and that neither sampling method was better than the other. In addition, both sampling methods showed high sensitivity and specificity. Our results are consistent with multiple studies showing that saliva sampling is just as sensitive as, or more sensitive than, OPS and NPS sampling for detection of asymptomatic or symptomatic SARS-CoV-2 infections in individuals, outpatients, and inpatients (9, 16–18). However, other studies have reported lower sensitivity for saliva sampling than for OPS and NPS (19, 20). In this study, self-administered saliva sampling was shown to be a very consistent and sturdy sampling method; the variation in the IC $C_T$ values of the saliva samples was low, despite each sample being collected by the

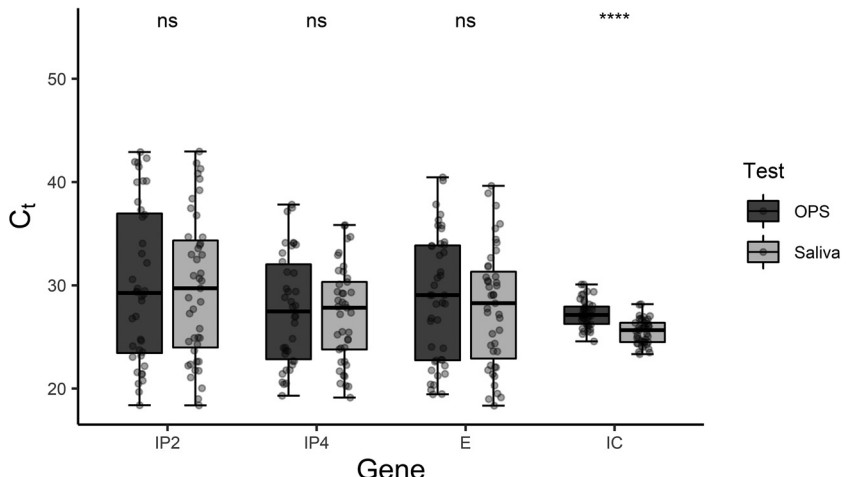

**FIG 3** Plot of $C_T$ values of three SARS-CoV-2-specific genes (IP2, IP4, and E) and the internal control (IC) for SARS-CoV-2-positive samples. The data are grouped by sampling method, oropharyngeal swab (OPS) ($n = 44$) or saliva ($n = 44$). Buffer 2 was used for preservation of the saliva. It can be seen that the mean $C_T$ values were similar for the oropharyngeal swab and saliva samples for the virus-specific genes, but a significant difference was found for the IC (an unpaired $t$ test was performed). See data in Table 3 posted at https://5ba.se/MLCetal/Saliva. ns, $P > 0.05$; *, $P \leq 0.05$; **, $P \leq 0.01$; ***, $P \leq 0.001$; ****, $P \leq 0.0001$.

participants themselves (Tables 3 and 4 posted at https://5ba.se/MLCetal/Saliva). The IC $C_T$ values of the saliva samples were significantly lower than those of the OPS samples for both the positive and negative samples (Tables 3 and 5 posted at https://5ba.se/MLCetal/Saliva).

Some of the side-by-side samples in this study showed discordance, which most likely is due to low virus titers in the samples, as seen for participants 11, 20, 28, and 43. However, participants 31 and 32 had relatively moderate viral signals in the OPS test ($C_T$, 28.5 to 29.8) and no viral signals in the saliva test at all (Table 3 posted at https://5ba.se/MLCetal/Saliva). Three samples were found to be SARS-CoV-2 negative using OPS but invalid using the saliva sampling method. It was observed that these three samples contained insufficient saliva material (less than 2 mL) and primarily consisted of buffer.

Most studies investigating the use of saliva to detect SARS-CoV-2 have been performed on adults, and the studies which have been performed on children have shown contradictory results and only had small cohorts (21, 22). Variation in the performance of saliva sampling for SARS-CoV-2 detection has previously been reported to be caused by inconsistent sampling procedures (clear saliva, deep-throat saliva, sputum, etc., collected by drooling, spitting, or coughing), different collection devices, viscosity of the samples, and inconsistent analyses (5, 22). Our study shows that the preservation buffer is very important for the stability of the SARS-CoV-2 RNA, as well as the IC DNA, and the type of buffer could potentially contribute to this variation between studies. In concordance with our findings, a study found that pure saliva itself prolonged SARS-CoV-2 RNA stability, regardless of storage at RT or after freeze/thaw cycles, which emphasizes that saliva can be collected in a simple container without the need for expensive preservation buffers or other supplements, e.g., for sample collection at home (23). However, Ott et al. also noticed a decrease in human RNase P (RNP) RNA when pure saliva was stored at RT (23). Bulfoni et al. found that the collection of saliva in a stabilizing solution helped homogenize the saliva as well as the preservation of the SARS-CoV-2 RNA, resulting in significantly lower $C_T$ values for the stabilized saliva samples than for the pure saliva samples (24). Furthermore, a buffer with inactivating properties presumably enhances the safety of handling saliva samples compared to handling active SARS-CoV-2 during transportation and analyzation.

There are several limitations to this study. First, the stability test was performed using a pool of samples from one participant only. However, the differences in RNA stability were prominent. Second, in our comparison between OPS and saliva samples preserved with buffer 2, our cohort was primarily composed of young adults (564/639 participants) who were enrolled across three boarding schools. No data on symptoms were collected, and thus, it is unknown if the participants testing positive were asymptomatic or symptomatic; however, the students were not allowed to be at the boarding schools if they showed symptoms, so the majority of participants should have been asymptomatic. Another important note about the cohort is that the majority of positive cases were either prior confirmed cases or close contacts of persons who tested positive for SARS-CoV-2. This explains the positive case rate of 7.4% in the cohort compared to the positive rate of approximately 0.2 to 0.5% in Denmark in the same period (25). Last, it is recommended by the manufacturer of the saliva collection tubes not to drink, eat, or smoke 30 min prior to the collection of samples; we did not control this among the participants.

**Conclusions.** This study shows that there is strong agreement between OPS and saliva sampling for SARS-CoV-2 detection. This was achieved using an appropriate preservation buffer; we found buffer 2 to be a much better choice for keeping the RNA stable in the saliva samples than buffer 1. In addition, both sampling methods had comparably high sensitivity and specificity. Self-administered saliva sampling is a promising candidate for controlling COVID-19 using large-scale testing without direct patient-health care personnel interaction. This study contributes to the evidence that saliva sampling is an attractive alternative to OPS for SARS-CoV-2 detection.

## MATERIALS AND METHODS

**Sample collection.** Sampling for the SARS-CoV-2 testing was performed with written consent from the participants using OPS and saliva samples collected side by side subsequently after each other. The participants were instructed not to drink, eat, or smoke within 30 min before the samples were collected; however, it was not documented whether the participants had followed these instructions. The test sampling was performed at the national test centers of PentaBase A/S (Denmark) and at Danish boarding schools.

Saliva was sampled via the drooling technique by the subjects themselves into saliva collectors (Biocomma Ltd., China). The subjects were supervised and instructed by trained personnel. Liquid saliva (2 mL) was collected using a collection funnel and mixed 1:1 with either the inactivating preservation solution buffers 1 or 2 (Biocomma Ltd.; catalog number SC12-C193 and SC11B, respectively) or left as pure saliva, depending on the experiment.

The oropharyngeal swab sampling was performed by trained personnel using a disposable swab (Biocomma Ltd.) and a sterile wooden disposable spatula (MaiMed GmbH, Germany) to fixate the tongue during the swabbing. Using firm pressure, rotational or painting motions were made with the swab on the mucosa at the posterior wall of the oropharynx and at the tonsils on both sides of the mouth. Touching the tongue and gums with the swab was avoided. The swab was snapped off at the breaking point into the testing tubes containing SARS-CoV-2-inactivating preservation buffer (Biocomma Ltd.; catalog number YMJ-TE3).

**Extraction and detection of SARS-CoV-2 RNA.** All samples were homogenized by vigorous vortexing for 1 min before purification of the RNA using a nucleic acid extraction kit (TianLong Science and Technology, China) with the automatic BasePurifier 32 nucleic acid extraction system (32 oscillating rods; PentaBase A/S; reference no. 715). All samples were purified in parallel with negative and positive controls for validation of the complete workflow. Purified RNA (5 $\mu$L) was transferred to the CoviDetect FAST COVID-19 multiplex RT-qPCR assay tubes (PentaBase A/S; reference number 8022) and analyzed using the BaseTyper 48.4 quiet high-resolution melting (HRM) real-time PCR system (PentaBase A/S; reference number 754). The RT-qPCR conditions consisted of an incubation period of 3 min at 52°C and a hold of 30 s at 95°C, followed by 45 cycles of 2-step amplification of 1 s at 90°C and 12 s at 60°C. Three SARS-CoV-2 viral sequences are targeted by the assay: two regions of the RNA-dependent RNA polymerase gene (here noted IP2 and IP4) and one region of the envelope protein gene (E). For validation of sampling and analysis, the presence of an internal control (IC), human RNase P (RNP) DNA, is measured. The limit of detection (LOD) of the assay has been determined to be 5 copies per reaction for both the OPS samples and the saliva samples (26).

**Data analysis—interpretation of results.** A sample can either be SARS-CoV-2 positive, negative, or invalid. The sample is positive for SARS-CoV-2 when at least two threshold cycle ($C_T$) values for the viral IP2, IP4, and E assays are lower than 41. Furthermore, a sample is also positive if only one out of the three virus-specific genes comes up with a $C_T$ value lower than 41 in two independent runs, given that all controls are valid. The sample is considered negative for detection of SARS-CoV-2 if the sample is positive for RNase P but negative for IP2, IP4, and E. In the case of no or late amplification of RNase P ($C_T$, ≥34), the test is invalid unless at least two out of three (IP2, IP4, and E) are positive ($C_T$, <41). The results

**TABLE 2** Cohort information

| Demographic | Data for participants | | |
| --- | --- | --- | --- |
| | Overall[a] | SARS-CoV-2 positive[b] | SARS-CoV-2 negative[c] |
| Male, $n$ (%) | 313 (49.0) | 27 (57.4) | 286 (48.3) |
| Age range (yrs [mean]) | 7–68 (25.5) | 7–59 (30.6) | 11–68 (25.1) |

[a]$n = 639$.
[b]$n = 47$.
[c]$n = 592$.

are only valid if the included positive-control $C_T$ values are ≤34 for IP2, ≤33 for IP4 and E, and ≤28 for the RNase P internal control. No template control (NTC) is also included in each run and should not produce $C_T$ values (26).

**Statistical methods.** To analyze the data and test for significance, statistical tests were performed using R version 4.1.1. An unpaired $t$ test was performed for comparison of the mean $C_T$ values of saliva stored in buffers 1 and 2 and the mean $C_T$ value of the OPS sampling for the positive samples. For comparison of saliva and OPS, a sample testing positive using only one method was included in the analysis. To test if the $C_T$ values of the OPS and saliva sampling methods were similar, a $t$ test was performed. The $t$ test was performed without pairing the samples, as some variance in $C_T$ values from each patient sample was expected. The $t$ test was used to measure whether the mean $C_T$ value for each gene varied significantly between the two sampling methods. A two-way analysis of variance (ANOVA) was performed for comparison of the mean $C_T$ value of the IC for the two sampling methods and the result of the SARS-CoV-2 analysis. A Tukey honestly significant difference (Tukey HSD) *post hoc* test was performed on the two-way ANOVA. Cohen's kappa value was calculated using the CohenKappa() function in the R package DescTools to measure the agreement between the two sampling methods (27). The sensitivity and specificity of the sampling methods were calculated as described by Baratloo et al. (28), and the confidence intervals were calculated using the Wilson score method.

**Specifications of each experiment. (i) SARS-CoV-2 RNA preservation in saliva using buffer 1.** The cohort comprised seven cases which had all been confirmed positive by OPS RT-PCR earlier the same day in February 2021. Both OPS and saliva samples were collected and then stored at 4°C. The saliva samples were mixed with buffer 1, whereafter RNA extraction and RT-PCR were performed.

**(ii) RNA stability in saliva and preservation media.** Six saliva samples were collected from an OPS RT-PCR-confirmed SARS-CoV-2-positive case, three of which were immediately stored at 4°C and the other three stored at room temperature (RT).

The three saliva samples kept at RT were pooled around 8 h after collection and split into three aliquots. One part was mixed 1:1 with buffer 1, one part was mixed 1:1 with buffer 2, and the last part was kept as pure saliva. The same procedure was used as for the three saliva samples kept at 4°C. Over a period of 9 days, SARS-CoV-2 detection was performed: four times in the first 2 days, followed by once a day for the next 3 days, and a final analysis on day 9. SARS-CoV-2 detection was conducted by performing RNA extraction for every RT-PCR analysis.

**(iii) Comparison of oropharyngeal swab and saliva sampling using buffer 2.** The samples were collected from 639 participants between March 2021 and June 2021. No data on symptoms were collected. The median age of the participants was 25.5 years, and 49.0% of them were male (Table 2). The saliva samples were mixed with buffer 2. All samples were processed within the same day. If either of the samples (OPS or saliva) belonging to the same individual were found to be SARS-CoV-2 positive, both samples were rerun in triplicate side by side, and the mean $C_T$ value of the three runs was calculated for each gene. The subjects testing positive for SARS-CoV-2 included both symptomatic and asymptomatic individuals. Some were close contacts of persons who tested positive for SARS-CoV-2 and some were cases confirmed previously via rapid antigen test or RT-PCR. Of the 639 tested subjects, 564 came from three different Danish boarding schools which had one or several of their students test positive for SARS-CoV-2 using a rapid antigen test.

## ACKNOWLEDGMENT

During the study, the authors worked at PentaBase A/S, Denmark, who supply RT-qPCR assays for detection of SARS-CoV-2 in OPS, NPS, or saliva samples.

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
