## [Reviewer comments · Microbiology Spectrum]

Microbiology Spectrum

Comparison of SARS-CoV-2 Detection from Saliva Sampling and Oropharyngeal Swab

Mia Clemmensen, Kamilla Bendixen, Katharina Flugt, Pernille Pilgaard, and Ulf Christensen

Corresponding Author(s): Mia Clemmensen, PentaBase A/S

Review Timeline:

Submission Date:	May 4, 2022
Editorial Decision:	May 26, 2022
Revision Received:	August 29, 2022
Accepted:	September 5, 2022

Editor: Miguel Martinez

Reviewer(s): Disclosure of reviewer identity is with reference to reviewer comments included in decision letter(s). The following individuals involved in review of your submission have agreed to reveal their identity: Yongbao wang (Reviewer #2)

Transaction Report:

DOI: <https://doi.org/10.1128/spectrum.01422-22>

May 26, 2022

Mx. Mia de Laurent Clemmensen
PentaBase A/S
Odense 5000
Denmark

Re: Spectrum01422-22 (Comparison of SARS-CoV-2 Analysis from Saliva Sampling and Oropharyngeal Swab)

Dear Mx. Mia de Laurent Clemmensen:

Your manuscript has been considered by two reviewers recruited for their expertise in the field. As their comments indicate, these individuals felt that your manuscript contained interesting observations but that it required modification before it could be considered acceptable for publication. Specifically, you should address the concerns raised by reviewer 1.

Link Not Available

Sincerely,

Miguel Martinez

Journals Department
Reviewer comments:

Reviewer #1 (Comments for the Author):

The manuscript "Comparison of SARS-CoV-2 Analysis from Saliva Sampling and Oropharyngeal Swab" aims to compare the SARS-CoV-2 molecular detection performance using two different specimens which present significant differences in terms of sample collection difficulty. The manuscript is interesting as all data that may contribute to the development of alternative methods that facilitate viral detection is relevant.

Specific comments on the manuscript:

- the title is not entirely correct. In fact, the manuscript does not deal with the analysis of SARS-CoV-2 but with its detection in two different matrices. As a suggestion the word "analysis" should be replaced by "detection";

- Line 19: "It was found that pure saliva kept the virus stable for 9 days...". Stability of a virus is usually measured by their viability to infect cells. The use of the word "stable" in the context of the work presented in the manuscript is therefore incorrect and this issue should be addressed all over the text;

- the introduction could be more balanced displaying more information regarding the core topic of the manuscript and less in areas that are not that relevant for the work that is presented. For instance, the authors present nothing less than 6 bibliographic references (7-12) concerning the Omicron variant and very few references concerning works that are already published and that are very similar to the one that is presented, i.e. testing the performance of saliva as a specimen to detect SARS-CoV-2 instead of NPS or OPS. An inversion of this situation would enrich the introduction;

- Line 89 - Buffers 1 and 2 are referenced using the catalog number but this is incomplete as the manufacturer name is not mentioned;

- Line 99 - the title of this section (2.2) is not adequate. It should be more explicit. The section should also have the PCR conditions that have been used because this is the basis of the work performed;

Line 100 - how was the sample homogenization performed?

Line 141 - Title of section 2.5.1 is not correct. The preservation of the virus SARS-CoV-2 is not the subject of the study. Otherwise the authors should report results on the infectivity of samples;

Lines 142-143 - the authors state that the "The cohort comprises seven cases which had all been confirmed positive by RT-PCR earlier the same day in February 2021." What type of specimens were used to confirm the samples positivity?

Line 144 - The authors state that "The saliva samples were mixed with Buffer 1 and RT-PCR was performed.". No RNA extraction was performed after mixing the samples with Buffer 1?

Line 147 - What type of specimens were used to confirm the samples positivity?

Lines 150-154 - the authors state that "Over a period of nine days, RT-PCR was performed...". Please clarify: during this period the each sample was subjected to several extractions? (one extraction for every RT-PCR analysis?);

Line 171 - Title of this section is not correct. The authors are not analysing the SARS-CoV-2 preservation. Should be corrected.

Line 183-185 - The authors state that "The Ct values for the SARS-CoV-2 specific genes were on average increased by 2.5 cycles for pure saliva stored at 4{degree sign}C and by 3.9 cycles when stored at RT during a time span of nine days.". However, in the material and methods section and also in the caption of figure 2, the authors refer to 3 specific genes (IP2, IP4 and E);

Line 209 and others - please clarify how was calculated the specificity;

Lines 240-242 - the phenomenon is presented as a bias but is perfectly normal and commonly observed in multiplex reactions; The manuscript deals with human samples but there are no reference to an approval of the work by an ethic commission nor the consent of the participants for the study.

Reviewer #2 (Comments for the Author):

well controlled study with two samples types, those conclusions for saliva based COVID testing from previous studies are mixed, this study indeed gave some very positive conclusion for saliva samples, this may relate to oropharyngeal swab, different from nasopharyngeal swab.

Staff Comments:

Preparing Revision Guidelines

Please return the manuscript within 60 days; if you cannot complete the modification within this time period, please contact me. If you do not wish to modify the manuscript and prefer to submit it to another journal, please notify me of your decision immediately so that the manuscript may be formally withdrawn from consideration by Microbiology Spectrum.

well controlled study with two samples types, those conclusions for saliva based COVID testing from previous studies are mixed, this study indeed gave some very positive conclusion for saliva samples, this may relate to oropharyngeal swab, different from nasopharyngeal swab.

this study will add knowledge about these two samples methods, it has merits on diagnostic testing.

Response to Reviewers

Reviewer #1:

The manuscript "Comparison of SARS-CoV-2 Analysis from Saliva Sampling and Oropharyngeal Swab" aims to compare the SARS-CoV-2 molecular detection performance using two different specimens which present significant differences in terms of sample collection difficulty. The manuscript is interesting as all data that may contribute to the development of alternative methods that facilitate viral detection is relevant.

Specific comments on the manuscript:

- the title is not entirely correct. In fact, the manuscript does not deal with the analysis of SARS-CoV-2 but with its detection in two different matrices. As a suggestion the word "analysis" should be replaced by "detection";

We agree with you. The title has been changed. All added words in the manuscript have been marked with yellow. All deleted words in the manuscript have been marked with red and strikethrough.

- Line 19: "It was found that pure saliva kept the virus stable for 9 days...". Stability of a virus is usually measured by their viability to infect cells. The use of the word "stable" in the context of the work presented in the manuscript is therefore incorrect and this issue should be addressed all over the text;

We see your point in line 19, as it was written as viral stability and not viral RNA stability. We have changed the wording of line 19 as well as headline 2.5.1 and 3.1. However, we have to disagree regarding the use of the word "stable" in the context of SARS-CoV-2 RNA stability as we find it suitable. Referring to the Medical Subject Headings (MeSH), the definition of RNA Stability is "The extent to which an RNA molecule retains its structural integrity and resists degradation by RNASE, and base-catalyzed HYDROLYSIS, under changing *in vivo* or *in vitro* conditions"¹, and we find it to fit the subject matter of the work.

- the introduction could be more balanced displaying more information regarding the core topic of the manuscript and less in areas that are not that relevant for the work that is presented. For instance, the authors present nothing less than 6 bibliographic references (7-12) concerning the Omicron variant and very few references concerning works that are already published and that are very similar to the one that is presented, i.e. testing the performance of saliva as a specimen to detect SARS-CoV-2 instead of NPS or OPS. An inversion of this situation would enrich the introduction;

We see your point about the balance of the core topic and the section about the Omicron variants. It was written in the first place to emphasize that SARS-CoV-2 testing is still important although the infectious rate at the time was not that high. Adjustments have been made to the introduction by removing some of the less relevant parts.

- Line 89 - Buffers 1 and 2 are referenced using the catalog number but this is incomplete as the manufacturer name is not mentioned;

Thank you for noticing. The manufacturer has been added.

- Line 99 - the title of this section (2.2) is not adequate. It should be more explicit. The section should also have the PCR conditions that have been used because this is the basis of the work performed;

¹ <https://meshb.nlm.nih.gov/record/ui?ui=D020871> (RNA Stability)

The title has been changed. Reference numbers has been added in the section for the product used from PentaBase A/S and the PCR conditions have been added.

Line 100 - how was the sample homogenization performed?

The method has been added: vigorous vortexing for 1 minute.

Line 141 - Title of section 2.5.1 is not correct. The preservation of the virus SARS-CoV-2 is not the subject of the study. Otherwise the authors should report results on the infectivity of samples;

The wording of section 2.5.1 has been changed by adding "RNA" in the headline.

Lines 142-143 - the authors state that the "The cohort comprises seven cases which had all been confirmed positive by RT-PCR earlier the same day in February 2021." What type of specimens were used to confirm the samples positivity?

Oropharyngeal specimen – it has been added.

Line 144 - The authors state that "The saliva samples were mixed with Buffer 1 and RT-PCR was performed.". No RNA extraction was performed after mixing the samples with Buffer 1?

Yes, the RNA was extracted after mixing the saliva with Buffer 1. It has been added.

Line 147 - What type of specimens were used to confirm the samples positivity?

Oropharyngeal swab – it has been added.

Lines 150-154 - the authors state that "Over a period of nine days, RT-PCR was performed...". Please clarify: during this period the each sample was subjected to several extractions? (one extraction for every RT-PCR analysis?);

Yes, it has been added. An RNA extraction for every RT-PCR analysis.

Line 171 - Title of this section is not correct. The authors are not analysing the SARS-CoV-2 preservation. Should be corrected.

The wording of section 3.1 has been changed by adding "RNA" in the headline.

Line 183-185 - The authors state that "The Ct values for the SARS-CoV-2 specific genes were on average increased by 2.5 cycles for pure saliva stored at 4{degree sign}C and by 3.9 cycles when stored at RT during a time span of nine days.". However, in the material and methods section and also in the caption of figure 2, the authors refer to 3 specific genes (IP2, IP4 and E);

Yes, it has been elaborated. $\Delta Ct = 2.5$, $\Delta Ct = 2.5$, and $\Delta Ct = 2.4$ for IP2, IP4, and E gene, respectively for pure saliva stored at 4°C giving an average of $\Delta Ct = 2.5$.

$\Delta Ct = 5.6$, $\Delta Ct = 2.4$, and $\Delta Ct = 3.6$ for IP2, IP4, and E, respectively for pure saliva when stored at RT giving an average of $\Delta Ct = 3.9$.

Line 209 and others - please clarify how was calculated the specificity;

It has been specified in section 2.4 now. The specificity and sensitivity have been calculated as described by Baratloo et al., "Part 1: Simple Definition and Calculation of Accuracy, Sensitivity and Specificity," Emergency, vol. 3, no. 2, pp. 48–49, 2015.²

However, your question got us to look at our calculations again, and we discovered a typing mistake. This mistake wrongly resulted in different specificities for the two sampling methods. It

² <https://www.ncbi.nlm.nih.gov/pmc/articles/PMC4614595/>

has now been corrected. In addition, the confidence intervals have been calculated anew using the Wilson Score method to consider the binomial distribution of the data, which we did not do before.

Lines 240-242 - the phenomenon is presented as a bias but is perfectly normal and commonly observed in multiplex reactions;

We agree. It has been deleted.

The manuscript deals with human samples but there are no reference to an approval of the work by an ethic commission nor the consent of the participants for the study.

The first line in the methods section mentions that written consent was obtained from each participant before performing the saliva sampling. Furthermore, our study was conducted prior to 26th May 2022. Thus, by Danish national law, the study did not need ethical approval as the study was non-invasive, and it was not an interventional clinical performance study.

Reviewer #2:

“well controlled study with two samples types, those conclusions for saliva based COVID testing from previous studies are mixed, this study indeed gave some very positive conclusion for saliva samples, this may relate to oropharyngeal swab, different from nasopharyngeal swab. this study will add knowledge about these two samples methods, it has merits on diagnostic testing.”

Thank you for your feedback – We appreciate it.

September 5, 2022

Mx. Mia de Laurent Clemmensen
PentaBase A/S
Odense 5000
Denmark

Re: Spectrum01422-22R1 (Comparison of SARS-CoV-2 Detection from Saliva Sampling and Oropharyngeal Swab)

Dear Mx. Mia de Laurent Clemmensen:

Your manuscript has been accepted, and I am forwarding it to the ASM Journals Department for publication. You will be notified when your proofs are ready to be viewed.

Sincerely,

Miguel Martinez
Editor, Microbiology Spectrum
